# Anesthesia Management in Sternal Resection for Chondrosarcoma: A Multidisciplinary Successful Approach with Thoracic Epidural and Parasternal Block for Acute and Long-Term Pain Control

**DOI:** 10.3390/jcm14228257

**Published:** 2025-11-20

**Authors:** Ouanes Amine Ben Saad, Benoit Rouiller, Corinne Grandjean, Sina Grogg, Jon Andri Lutz, Chloe Mimouni

**Affiliations:** 1Service of Anesthesiology and Reanimation, Fribourg Hospital, 1708 Fribourg, Switzerland; 2Thoracic Surgery Unit, Department of Surgery, Fribourg Hospital, 1708 Fribourg, Switzerland

**Keywords:** chondrosarcoma, regional anesthesia, thoracic anesthesia, thoracic surgery, chest wall resection, long-term pain control

## Abstract

**Background:** Although rarely performed, sternal resection for chondrosarcoma presents considerable anesthetic challenges, particularly in both immediate and long-term pain management. **Method:** This case study details the anesthetic protocol adopted for a 61-year-old male who underwent a sternal resection, chest wall reconstruction and muscle flap coverage due to chondrosarcoma. To optimize perioperative analgesia, a thoracic epidural catheter was placed preoperatively, combined intraoperatively with bilateral parasternal blocks targeting intercostal nerves at the resection margin. General anesthesia was tailored and paired with fluid restriction and minimal vasopressor doses to stabilize hemodynamics. **Result:** Postoperative recovery was marked by minimal discomfort, no need for systemic opioids, and early initiation of physiotherapy. At 12 months post-surgery, the absence of persistent pain or physical dysfunction suggested that the synergistic use of thoracic epidural and parasternal blocks offers effective control over both short-term and chronic pain following major chest wall surgery. The holistic integration of psychological support and an active rehabilitation regimen exemplified a biopsychosocial strategy, instrumental to the patient’s positive trajectory. **Conclusions:** This experience underscores the advantage of supplementing epidural anesthesia with parasternal nerve blocks in sternal resections, facilitating better outcomes and reducing systemic opioid use. Further studies involving broader patient populations are necessary to validate and refine this promising approach in preventing chronic pain in complex thoracic surgeries.

## 1. Introduction

Primary tumors of the chest wall constitute 1 to 2% of all thoracic neoplasms. Approximately 60% are malignant, most commonly sarcomas, with around 55% originating from bone or cartilage and about 45% from soft tissue components. This case report presents the anesthesia management of a 61-year-old patient undergoing sternal resection for chondrosarcoma, a rare and complex thoracic surgery. Given the significant postoperative pain associated with this procedure, we employed a multimodal approach, including a thoracic epidural for acute pain and a parasternal block, hypothesized to have contributed to improved long-term pain control. Both perioperative and follow-up data suggest that this combination was highly effective in managing both acute and chronic post-surgical pain. Sternal resections are rare surgical procedures, typically performed for conditions like chondrosarcoma, which represents a challenging area of perioperative anesthesia management. Pain control in such cases is critical, given the invasive nature of the surgery with high perioperative morbidity and the high potential for both acute and chronic post-surgical pain. This report focuses on a novel approach combining thoracic epidural analgesia with a parasternal block to optimize both short-term and long-term pain outcomes, while also emphasizing the importance of a biopsychosocial approach in managing the patient’s overall well-being.

## 2. Case Presentation

### 2.1. Patient Background

A 61-year-old highly educated man, with a 20 pack-year smoking history, hypertension, and benign prostatic hyperplasia, presented with a of a year history of persistent dull sternal pain and progressive swelling. Initial CT (Figure 1) and MRI revealed calcified mass involving most of the sternal body, invading pre-pectoral muscles and displacing pericardial fat and pleurae without direct invasion. CT-guided biopsy confirmed chondrosarcoma, and PET imaging demonstrated intense uptake at the primary site without metastasis. As complete resection with clear margins was deemed feasible, in accordance with the literature [1], upfront surgery with chest wall reconstruction was proposed. Despite experiencing anxiety and low mood at diagnosis, the patient was motivated to proceed despite anticipating postoperative pain.

### 2.2. Surgical Procedure

Thoracic and plastic surgeons successfully performed an *en-bloc* resection, which included the sternal skin patch, underlying pectoral musculature, the corpus of the sternum, and the costal cartilages of ribs 2 through 6 bilaterally. The adjacent pleura and mediastinal fat were also resected. Reconstruction was achieved using five transverse titanium rib fixation plates, covered with a polypropylene mesh. The pectoral musculature was reattached to the mesh (Figure 2). A left latissimus dorsi musculocutaneous flap was tunneled subcutaneously and secured to both the mesh and the remaining pectoral musculature, following confirmation of its vascular integrity (Figure 3).

### 2.3. Anesthesia Management

Given the invasiveness of the procedure and the patient’s anticipated postoperative pain and anxiety, premedication was administered 30 min prior to anesthesia with Lorazepam 1mg and slow-release Oxycodone 10 mg. The patient underwent anesthesia and surgery with continuous monitoring, including invasive blood pressure, pulse oximetry, electrocardiogram, train-of-four, and bispectral index (BIS). A thoracic epidural was placed preoperatively at the T5–T6 level for intra- and postoperative pain control, followed by general anesthesia (GA) induced intravenously using Propofol and sufentanil. Rocuronium was administered to maintain deep muscle relaxation. Intubation was performed using a video-laryngoscope to minimize risks associated with difficult airway management. GA was maintained with target-controlled infusion of Propofol and boluses of sufentanil (with BIS monitor between 40 and 60), with a continuous infusion of a mixture of Bupivacaine 0.25% and Fentanyl 2.5 mcg/mL through the epidural catheter at 8 mL/h. A central venous catheter was placed after intubation in anticipation of intraoperative hemodynamic instability and the potential need for vasopressors. To avoid fluid overload, we employed a restrictive fluid therapy strategy, with noradrenaline infused as needed to maintain hemodynamic stability. The patient received 5 mL/kg/h of crystalloid fluid and a mean of 0.02 mcg/kg/min of noradrenaline. As part of the multimodal analgesia management, paracetamol 1 g was administered twice during GA, along with a 50 mg/kg bolus of magnesium. Additionally, a parasternal block was administered bilaterally intraoperatively by the surgeons under direct vision, targeting the intercostal nerves adjacent to the resected sternum with ropivacaine 0.5% (100 mg total). The anesthesia and surgery were successful and uneventful, with the patient remaining hemodynamically stable throughout. The procedure was completed in 7 h and 13 min, with general anesthesia lasting 8 h and 32 min. The patient was transferred intubated and sedated to the ICU for postoperative care.

### 2.4. Post-Operative Care

The patient was extubated after 48 h and discharged from the ICU after removal of bilateral chest drains on postoperative day (POD) 5. During the ICU stay, the patient reported minimal pain in the immediate postoperative period, with a numeric rating scale (NRS) of 2/10 at rest and on movement. In the surgical ward, the combination of the epidural (continuous with bolus available every 45 min) and parasternal block provided excellent analgesia without the need for systemic opioids. The epidural was gradually weaned over 7 days, then gradually tapered before removal with a transition to per-os slow-release oxycodone 10 mg, around-the-clock paracetamol 1 g, and metamizole 1 g, with rapid-release oxycodone 5 mg as a backup. Total opioid consumption during the first 72 h was markedly lower (20 mg oxycodone) than typical thoracic resection series reporting > 300 morphine milligram equivalent [2]. Early physiotherapy sessions were initiated for analgesic and anti-inflammatory purposes, as well as to improve joint and muscle function. The slow-release oxycodone was weaned over 4 days, and the patient was discharged on POD 12 with over-the-counter WHO step-1 analgesics and physiotherapy sessions (3×/week). Postoperative surgical follow-up was unremarkable, except for a seroma in the area of the latissimus dorsi muscle flap, which resolved spontaneously.

### 2.5. LongTerm Follow-Up

Follow-up appointments at 1, 3, 6, and 12 months were reassuring, with no signs of relapse or complaints from the patient. The first follow-up scan at 6 months showed no evidence of recurrence. At the 1-year follow-up (Figure 4), the patient appeared well, with good posture, no gait abnormalities, and no musculoskeletal deformities, residual swelling, or muscle atrophy. The skin scar was slightly retracted but without allodynia. Palpation revealed no tenderness, trigger points, muscle spasms, or joint effusion. Spinal and thoracic range of motion were full, with no limitations. Strength testing and neurological examination were normal, with no evidence of chronic or neuropathic pain.

## 3. Discussion

Sternal resection for malignancies such as chondrosarcoma is rare and poses significant anesthetic challenges. Thoracic epidural analgesia remains the standard for intra- and postoperative pain control, but chronic pain remains a concern, affecting up to 50% of patients after major thoracic surgery [3]. In this case, we combined a thoracic epidural with a parasternal block targeting the anterior, lateral, and posterior cutaneous branches of the intercostal nerves. This approach aimed to prevent chronic pain by interrupting nociceptive input from the resected sternum. 

Nerve entrapment syndromes following thoracotomy can lead to chronic pain and autonomic disturbances, including altered vasomotor control, dyshidrosis, and visceral-somatic pain misinterpretation [4]. The parasternal block was selected based on cadaveric studies demonstrating good intercostal diffusion and our institutional experience in minimally invasive chest wall procedures [5]. 

While epidurals effectively reduce acute post-thoracotomy pain, their impact on chronic pain is less clear [6]. Unlike PECS, SAPB, or ESP blocks that target the lateral chest wall, the parasternal block directly covers the anterior intercostal branches, complementing epidural dermatomal coverage. This targeted blockade may reduce local inflammation and prevent central sensitization by limiting nociceptive afferent input, thus avoiding dorsal horn hyperexcitability and maladaptive plasticity [7]. This mechanism could explain the absence of chronic pain at 12 months, supporting the concept that early and comprehensive nociceptive blockade can mitigate persistent post-surgical pain [6].

The combined epidural–parasternal technique provides analgesia for both deep (periosteal and muscular) and superficial (cutaneous) components of the sternal wound, potentially improving coverage compared with lateral plane blocks. To our knowledge, this is among the first reports describing this combination in extensive sternal resection, suggesting a logical and potentially superior option for anterior thoracic surgeries [4,5]. Our patient’s minimal pain at 12 months is encouraging, given that up to half of sternotomy patients experience persistent pain [3,8].

The choice of thoracic epidural was based on its proven efficacy for major thoracic surgery [9]. Magnesium was administered for its analgesic and opioid-sparing effects and its potential role in modulating neuropathic and cancer-related pain via NMDA receptor and calcium channel interactions [10]. Restricted fluid therapy, guided by hemodynamic monitoring, aimed to minimize pulmonary edema and optimize respiratory function [11,12].

The surgical technique itself likely contributed to the positive outcome. The *en-bloc* resection with wide margins, followed by careful reconstruction with titanium plates and muscle flap coverage, aimed to provide both oncological control and structural stability [13].

Beyond technical factors, the patient’s psychological and social well-being were integral to success. Applying a biopsychosocial approach, preoperative anxiety and low mood were addressed through communication, reassurance, and psychological support, fostering resilience and cooperation. Early physiotherapy and structured rehabilitation promoted physical and psychological recovery, while the patient’s high level of education facilitated informed decision-making and adherence to care.

### Limitations

This is a single case, and causality between the parasternal block and the favorable outcome cannot be established. Psychological factors, adherence to physiotherapy, and meticulous surgical technique likely contributed. Quantitative sensory testing (QST) was not performed, limiting objective pain assessment. Larger studies with QST evaluation are needed to confirm the parasternal block’s role in preventing chronic post-sternotomy pain.

## 4. Conclusions

This case demonstrates the successful anesthesia and analgesia management of a complex sternal resection for chondrosarcoma using a multimodal technique combining thoracic epidural and parasternal block, together with a long-term multidisciplinary and biopsychosocial approach. The patient experienced minimal acute and chronic pain, suggesting that a parasternal block may be valuable adjunct to epidural analgesia in preventing long-term pain after sternal resection, while minimizing systemic opioid use. A multidisciplinary team approach, incorporating psychological support and early rehabilitation, is invaluable in pain management of chest wall tumors and reconstruction of chest wall defects. This tailored anesthetic strategy, along with early rehabilitation and restrictive fluid therapy, contributed to a favorable postoperative outcome. Further studies are warranted to explore the long-term benefits of this approach in preventing chronic post-sternotomy pain.

## Figures and Tables

**Figure 1 jcm-14-08257-f001:**
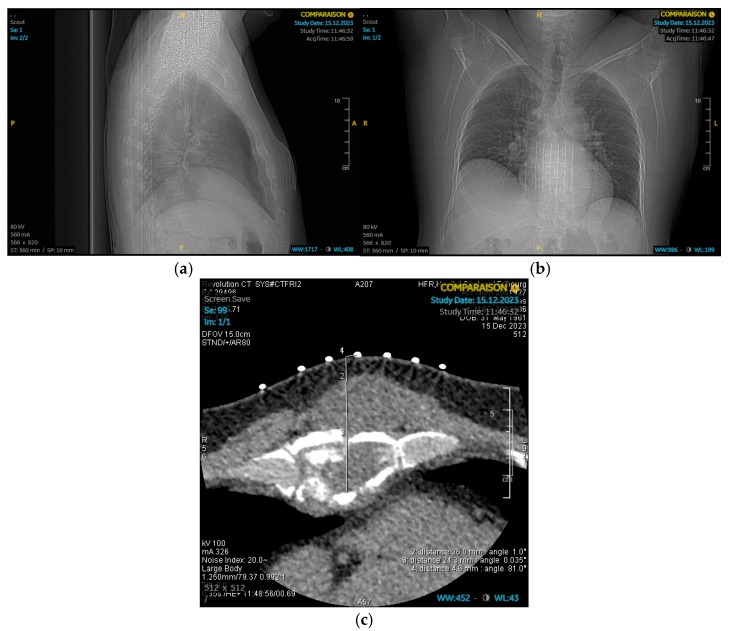
Chest CT. (**a**) Chest CT lateral view. (**b**) Chest CT posteroanterior view. (**c**) Chest CT slice: A mass with calcified components involving a large part of the sternal body.

**Figure 2 jcm-14-08257-f002:**
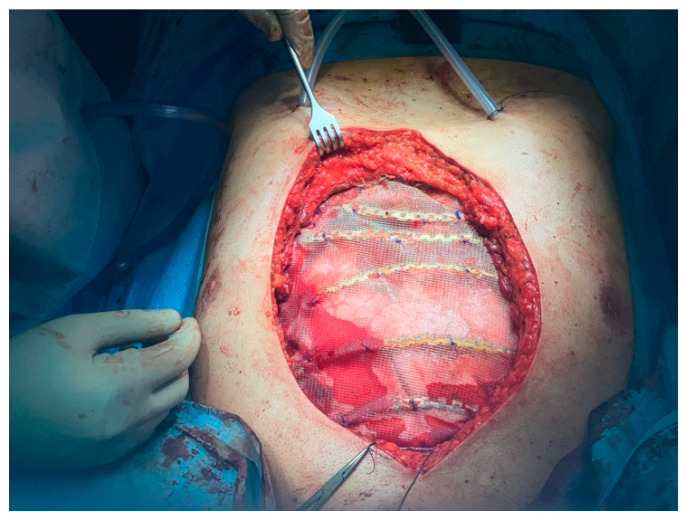
*en-bloc* resection and reconstruction with titanium rib fixation plates and covered with a polypropylene mesh. The block was performed bilaterally under direct surgical vision using 20 mL of Ropivacaine 0.5% (total dose = 100 mg), injected along the lateral borders of the sternum between the 2nd–6th costal cartilages.

**Figure 3 jcm-14-08257-f003:**
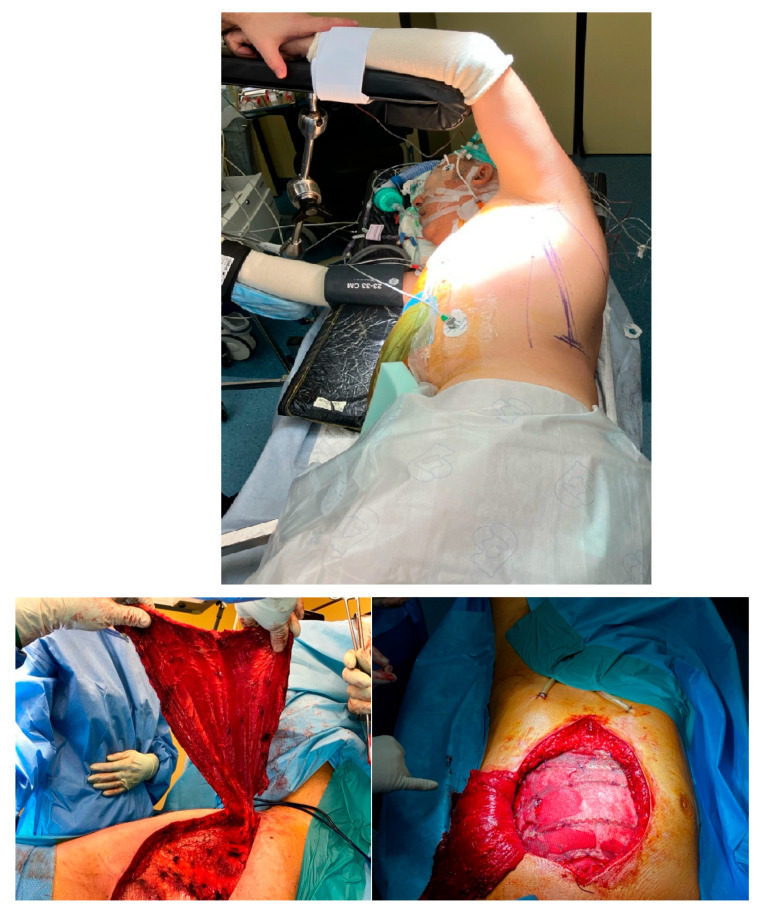
Left latissimus dorsi musculocutaneous flap.

**Figure 4 jcm-14-08257-f004:**
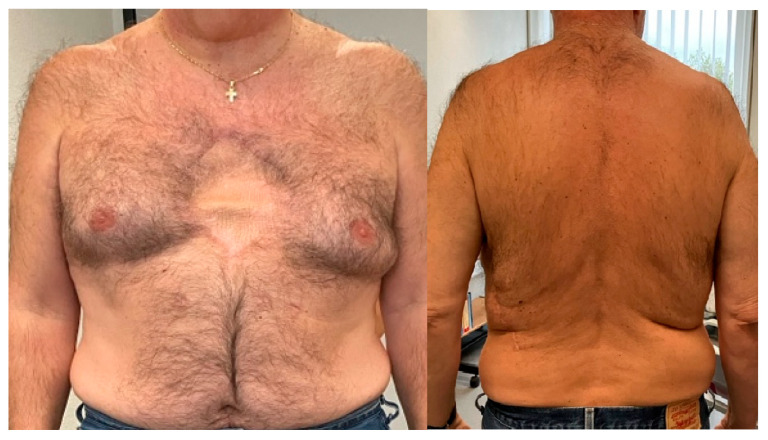
One-year follow-up.

## Data Availability

All relevant data are within the manuscript.

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
