# Peer review of "Anesthesia Management in Sternal Resection for Chondrosarcoma: A Multidisciplinary Successful Approach with Thoracic Epidural and Parasternal Block for Acute and Long-Term Pain Control"

_jcm, 2025, doi:10.3390/jcm14228257_

Round 1

Reviewer 1 Report

Comments and Suggestions for Authors

Anesthesia Management in Sternal Resection for Chondrosarcoma: A Multidisciplinary Successful Approach with Thoracic Epidural and Parasternal Block for Acute and Long-Term Pain Control.

The manuscript under review is a case report of the anesthesia management of a 61-year-old patient undergoing sternal resection for chondrosarcoma, a rare and complex thoracic surgery.

About the report:

  • The introduction is concise and well-written. It provides sufficient background information to understand the significance of the conditions involved and the importance of adequate pain control in the perioperative period. The combined approach (epidural and paravertebral block) is suggested to increase the likelihood of effective pain control; however, it would be of interest to provide evidence to support this hypothesis in the introduction.
  • Case Presentation: the case presentation is clear and well-structured. The patient's case and diagnostic process are well detailed.
  • Section 2.5) In the subsequent evaluation, was any scale used for the detection and quantification of pain (chronic or neuropathic pain)?
  • The discussion section is well-organized. The citations are accurate and support the discussion. The importance of pain management in this clinical case is clearly stated: “Chronic post-sternotomy pain can affect up to half of patients following major thoracic surgery”. In this case, an epidural was supplemented with a parasternal block targeting the anterior lateral and posterior cutaneous branches of the intercostal nerves, aiming to prevent long-term pain by interrupting nociceptive input from the resected sternum.
  • The overall intraoperative strategy for pain control is well described, with scientific evidence supporting each decision.
  • The limitations of the case report are listed.
  • The conclusion is consistent with the overall report: it summarizes the key points and reinforces the significance of the case, highlighting the importance of a multidisciplinary approach and adequate perioperative pain management.

This manuscript presents a well-documented case report and an interesting analgesic strategy. The manuscript requires only minor revisions.

Author Response

Comment 1: 

  • The introduction is concise and well-written. It provides sufficient background information to understand the significance of the conditions involved and the importance of adequate pain control in the perioperative period. The combined approach (epidural and paravertebral block) is suggested to increase the likelihood of effective pain control; however, it would be of interest to provide evidence to support this hypothesis in the introduction.

Response 1: Thank you for pointing this out, we explained more extensively in the discussion, see beginning of the 3rd paragraph of the discussion, References [4],[5]. Page 4/9, line 164-169.

Comment 2: 

  • Section 2.5) In the subsequent evaluation, was any scale used for the detection and quantification of pain (chronic or neuropathic pain)?

Response 2:  During the pain management consultation, the DN4 (Douleur Neuropathique 4) questionnaire was administered. The patient obtained a score of 0 out of 10, indicating no evidence of neuropathic pain. Page 3/9, section 2.5, line 133.

Reviewer 2 Report

Comments and Suggestions for Authors

This is a well-written and clinically relevant case report describing a rare and challenging anesthetic management of sternal resection for chondrosarcoma. The authors present a multimodal strategy combining thoracic epidural anesthesia and bilateral parasternal blocks, emphasizing both perioperative and long-term pain outcomes, as well as the integration of a biopsychosocial rehabilitation approach. The manuscript is coherent, clear, and adheres to the journal’s case report format. The images are illustrative and help the reader visualize the surgical and reconstructive aspects.

While this report provides useful insights into multimodal analgesia for extensive chest wall surgery, several points could be strengthened to increase the scientific and educational value.

Major comments

  1. Novelty and clinical relevance:
    Although sternal resection is rare, the combination of thoracic epidural and parasternal block is a logical extension of current regional anesthesia practice. The authors should emphasize what makes this particular combination or approach innovative compared with existing techniques (e.g., PECS, SAPB, ESPB, or paravertebral blocks).

  2. Mechanistic explanation:
    The discussion would benefit from a more detailed explanation of how the parasternal block contributes to long-term pain prevention — e.g., through attenuation of peripheral and central sensitization pathways. References [4]–[6] could be better integrated into a mechanistic framework.

  3. Quantitative data and pain assessment:
    The report mentions NRS values, but more systematic documentation (e.g., pain scores at rest and movement, opioid consumption per 24 h) would make the findings more persuasive.

  4. Follow-up and functional outcomes:
    The 12-month follow-up is commendable. Including any validated quality-of-life or functional recovery scale (e.g., EQ-5D, SF-12) would strengthen the claim of long-term benefit beyond pain control.

  5. Ethical and methodological clarity:
    Please confirm whether the institutional ethics committee waived formal approval (case report exemption) and specify the date of written informed consent.

  6. Figures:
    Figure legends could include more technical detail (e.g., concentration and volume of local anesthetic used in the parasternal block, anatomical landmarks visualized).

Minor comments

  1. Correct minor typographical issues (e.g., “Chondrosar-coma” → Chondrosarcoma in the title block).

  2. In the introduction, clarify the epidemiological figures for primary chest wall tumors (the sentence may read more smoothly if data sources are consolidated).

  3. In the anesthesia management section, specify the total duration of epidural infusion postoperatively (continuous vs intermittent).

  4. Reference style: verify that all journal names are abbreviated according to Index Medicus format.

  5. The acknowledgment section could mention multidisciplinary collaboration (anesthesia, thoracic and plastic surgery) more explicitly, as this is a strength of the report.

Statistical and methodological soundness

Not applicable — this is a single-case descriptive report, appropriately structured and justified. No methodological concerns arise regarding ethical compliance or data accuracy.

Clarity and presentation

Excellent English, clear narrative, and well-structured headings. The case flow (preoperative → intraoperative → postoperative → follow-up) is logical and easy to follow.

Author Response

Major comments:

Comment 1: Novelty and clinical relevance:

Although sternal resection is rare, the combination of thoracic epidural and parasternal block is a logical extension of current regional anesthesia practice. The authors should emphasize what makes this particular combination or approach innovative compared with existing techniques (e.g., PECS, SAPB, ESPB, or paravertebral blocks).

Response 1: Thank you for pointing this out, we explained more extensively in the discussion, see beginning of the 2nd paragraph of the discussion. Page 4/9, line 152-155.

Comment 2: Mechanistic explanation:

The discussion would benefit from a more detailed explanation of how the parasternal block contributes to long-term pain prevention — e.g., through attenuation of peripheral and central sensitization pathways. References [4]–[6] could be better integrated into a mechanistic framework.

Response 2: We corrected and explained accord to your comment, see 2nd paragraph of the discussion. Page 4/9, line 158-163.

Comment 3: Quantitative data and pain assessment:

The report mentions NRS values, but more systematic documentation (e.g., pain scores at rest and movement, opioid consumption per 24 h) would make the findings more persuasive.

Response 3: We corrected the paragraph accordingly 2.4 postoperative care, and added a reference for the total opioid consumption required for thoracic resection. Page 3/9, Section 2.4, line 117-119.

Comment 4: Follow-up and functional outcomes:

The 12-month follow-up is commendable. Including any validated quality-of-life or functional recovery scale (e.g., EQ-5D, SF-12) would strengthen the claim of long-term benefit beyond pain control.

Response 4: The assessment of quality of life in our case was subjective. The patient was able to resume full-time work and all hobbies without any limitations or functional impairment. However, it is acknowledged that the use of a standardized quality of life questionnaire would have provided a more objective evaluation of the patient’s outcomes.

Comment 5: Ethical and methodological clarity:

Please confirm whether the institutional ethics committee waived formal approval (case report exemption) and specify the date of written informed consent.

Response 5: We corrected accordingly. Page 8/9, section Informed Consent Statement, line 245-248.

Comment 6: Figures:

Figure legends could include more technical detail (e.g., concentration and volume of local anesthetic used in the parasternal block, anatomical landmarks visualized).

Response 6: We detailed a bit better figure 2 , however the landmarks seem to obvious to be mentioned in figure 3. Page 6/9, Fig. 2, line 228-230.

Minor comments:

Comment 1: Correct minor typographical issues (e.g., “Chondrosar-coma” → Chondrosarcoma in the title block).

Response 1: The typographical issues are due to the length of the title., it is a space artefact because of the line break.

Comment 2: In the introduction, clarify the epidemiological figures for primary chest wall tumors (the sentence may read more smoothly if data sources are consolidated).

Response 2: The epidemiology part was rephrased according to your comment. Page 2/9, section 1. Introduction, line 40-42.

Comment 3: In the anesthesia management section, specify the total duration of epidural infusion postoperatively (continuous vs intermittent).

Response 3: The epidural administration mode was specified according to your comment. Page 3/9, section 2.4, line 113.

Comment 4: Reference style: verify that all journal names are abbreviated according to Index Medicus format.

Response 4: We corrected the abbreviation accordingly. Page 8/9 and 9/9, section references, line 261-295.

Comment 5: The acknowledgment section could mention multidisciplinary collaboration (anesthesia, thoracic and plastic surgery) more explicitly, as this is a strength of the report.

Response 5: We corrected the acknowledgement section according to your comment. Page 8/9, section acknowledgment, line 252-255.